# Usage and Microbial Safety of Shared and Unshared Excreta Disposal Facilities in Developing Countries: The Case of a Ghanaian Rural District

Peter Appiah Obeng [1,*], Eric Awere [2,*], Panin Asirifua Obeng [2], Michael Oteng-Peprah [1], Albert Kaabieredomo Mwinsuubo [1,3], Alessandra Bonoli [4] and Sharon Amanda Quaye [5]

1. Department of Water and Sanitation, University of Cape Coast, P.M.B. University Post Office, Cape Coast, Ghana; moteng-peprah@ucc.edu.gh (M.O.-P.); albert.mwinsuubo@stu.ucc.edu.gh (A.K.M.)
2. Department of Civil Engineering, Cape Coast Technical University, Cape Coast P.O. Box DL 50, Ghana; asirifua.obeng@cctu.edu.gh
3. Environmental Health and Sanitation Management Department, Ajumako-Enyan-Essiam District Assembly, Ajumako P.O. Box 1, Ghana
4. DICAM Department, University of Bologna, via Terracini 28, 40131 Bologna, Italy; alessandra.bonoli@unibo.it
5. Independent Researcher, Accra P.O. Box CT-2912, Ghana; sharon.oboateng@gmail.com
* Correspondence: pobeng@ucc.edu.gh (P.A.O.); eric.awere@cctu.edu.gh (E.A.)

**Abstract:** Sharing facilities with other households offers the most realistic opportunity for access to sanitation for many households in low-income settings. However, questions remain about the safety of shared toilets, including those shared at the household level. This study sought to compare the usage and microbial safety of household-level shared and unshared toilets in a Ghanaian rural district to investigate any association between their microbial safety and sharing status. A semi-structured questionnaire was used to collect data on the user characteristics of the sampled toilets, while common contact surfaces (door handles and toilet seats) were assessed for faecal contamination following standard swab sampling and analytical protocols. The results of the study indicate that sharing toilets affords about 90% more household-level access to sanitation as compared to single-household toilets. Toilet sharing mostly occurred between two households, with a maximum user population of 14 per toilet. Generally, there was a high prevalence of faecal contamination on the door handles and seats of both shared and unshared toilets, but this had no association with the sharing status of the toilets. The median concentration of *Escherichia coli* (*E. coli*) on the door handles and seats of shared toilets was $34.3 \times 10^5$ and $103.2 \times 10^5$ CFU/mL, respectively, as compared to $54.7 \times 10^5$ and $125.0 \times 10^5$ CFU/mL, respectively, on unshared toilets. In conclusion, the sharing of toilets at the household level nearly doubles access to sanitation at home without necessarily exposing the users to a higher risk of faecal–oral disease transmission.

**Keywords:** shared sanitation; microbial safety; toilet sharing; toilet usage; Ghana; SDG 6.2; excreta disposal facilities

## 1. Introduction

Access to safe excreta disposal systems poses a significant challenge to low-income households in developing countries, especially those in rural areas. Available data from the World Health Organization (WHO) and the United Nations Children's Fund (UNICEF) [1] reveal that an estimated 21% (about 1.7 billion people) of the global population lacked at least basic sanitation services (individual household toilets) in 2020. The proportion varied across the Sustainable Development Goal (SDG) regions, with the highest proportion occurring in sub-Saharan Africa (68%).

For many of such households, sharing facilities with other households offers the most realistic alternative to unsafe communal facilities or the environmentally risky practice of open defecation. Some households may never be able to own individual toilets for

many reasons. These include poverty, which denies some households adequate financial resources to construct their own toilets, and issues of land ownership and tenure security. In sub-Saharan countries, for instance, tenants and settlers are discouraged from building private toilets on land that can be legally or traditionally claimed by others [2]. Another factor which discourages ownership of individual household toilets is multi-family housing (commonly called 'compound houses'), which exists in both rural and rapidly expanding cities in Africa. These housing units are shared by the generational extended family members or are rented to strangers in the family-dominated compounds. Householders in many of these housing units (between 20 to 200 people) share a living space and utilities such as water, electricity, and toilets [3]. Sharing sanitation facilities with extended family members and neighbours has been found to be more acceptable in certain cultures. Obeng et al. [4] found that over 80 percent of households in Prampram, Ghana, who had no toilets were either tenants (16 percent) or occupants of family houses (65 per cent) that were shared by several family or household units. Poor tenants with little bargaining power are unable to demand household toilets for fear of ejection [5].

However, from the Millennium Development Goals era, the sharing of facilities by more than one household was classified by the Joint Monitoring Programme (JMP) of the WHO and UNICEF as unimproved. The position of the JMP was informed by concerns associated with the hygiene, accessibility, and safety of shared facilities for users [2]. In the run-up to the formulation of the SDGs, the Sanitation Task Team (STT) convened by the JMP to advise on the targets and indicators for global monitoring recommended that sanitation facilities shared by up to five families and no more than 30 persons should be included in 'basic' sanitation [6]. Nevertheless, the JMP finally decided to exclude shared sanitation from the normative definitions for 'basic' and 'safely managed' sanitation with the explanation that it is practically challenging to distinguish poorly designed and managed shared facilities from those that are hygienic, accessible, and safe [6]. The exclusion of household-level shared sanitation facilities from, at least, the basic sanitation rung of the JMP's sanitation ladder has the potential to discourage governments and non-governmental organisations (NGOs) who are driven by international criteria for the assessment of progress towards the SDGs from paying attention to the needs of such households in the design of intervention packages.

In spite of the JMP's position, available published data show a varied opinion about the state of shared and unshared sanitation facilities. Some systematic reviews of health outcomes [7,8] and analysis of Demographic and Health Surveys (DHS) [9] found that the use of shared sanitation is associated with an increased risk of diarrhoeal diseases. Poor health outcomes and exposure to violence have been associated with unhygienic conditions and the inaccessibility of shared toilets. However, other studies found shared toilets to be comparable to individual household toilets in terms of health outcomes, faecal contamination, accessibility, and cleanliness. Montgomery et al. [10] found no difference in the risk of trachoma between households using shared or private toilets in rural Tanzania. In Mozambique, The Gambia, Rwanda, Senegal, and South Africa, the use of shared sanitation showed a protective effect against diarrhoea [11]. Evidence from Dar es Salaam, Tanzania, shows that shared toilets were positively associated with hygienically safe and functionally sustainable toilets [12]. Moreover, Exley et al. [13] found no evidence that shared sanitation facilities were more contaminated with *E. coli* than unshared toilets. Gunther et al. [14] found the cleanliness of toilets shared by 2–3 households to be comparable to that of unshared toilets in Kampala. Other reviews reported mixed findings. For instance, Obeng et al. [11] reviewed the literature on the vulnerabilities associated with the use of shared toilets in sub-Saharan Africa and concluded that there seems to be a varied opinion among experts on issues regarding the sharing of sanitation facilities. While the study found sharing sanitation to be a risk factor for non-partner violence against women and diarrhoeal diseases, it also found evidence that many shared facilities, particularly those shared by two or three households, are clean and afford the users similar health outcomes to non-shared facilities. These findings suggest that toilets that are not hygienic may be caused by factors

including poor user behaviour and ineffective management practices irrespective of the sharing status.

The JMP itself recognises the unavailability of adequate data to make a firm decision on shared sanitation. Furthermore, the debate over the safety of shared sanitation facilities is complicated by the fact that most existing works fail to segregate household-level shared facilities, such as those that meet the benchmark recommended by the STT, from those that are shared at the communal level. For instance, studies such as Ramlal et al. [7] and Heijnen et al. [8], among others, lumped communal/public toilets together with household-level shared ones and drew conclusions that could be significantly influenced by the communal toilets among the shared facility cohort. Quite often, reference is made to shared facilities in a sense that may be applicable to only communal or public toilets. For instance, when the JMP raises issues about poor design, unsafe management, lack of accessibility, etc., it appears to be in reference to communal or public toilets. Otherwise, it is not clear as to how that becomes an issue only with toilets that are shared by a few people from, say, cotenant households on their compound, but not a similar toilet used by a single household within the same socio-cultural setting. In other words, if communal facilities are segregated, the issues raised by the JMP may be recognised as matters of sanitation facility usage that are probably associated with the general awareness of and commitment to hygiene among the inhabitants of a particular socio-cultural setting rather than the sharing of facilities among a few people from two to five households as recommended by the STT.

There is, therefore, a need for more studies that compare the qualities of household-level shared facilities to those of facilities used by single households within the same geographical and socio-cultural context. Such studies are needed to generate more data to inform the discourse on the opportunities and threats offered by household-level shared facilities in attaining the global goals for access to sanitation. This paper seeks to contribute to this discourse by comparing the usage and microbial safety of facilities that are used by single households to those shared by multiple cotenant households in a Ghanaian rural district.

## 2. Materials and Methods

### 2.1. Study Area

The study was conducted in selected communities in the Ajumako–Enyan–Essiam District (AEED) of the Central Region of Ghana. The District is located between latitudes 5°53′ and 1°34′ north and longitudes 0°53′ and 1°08′ west [15]. Administratively, the District is structured into nine Area Councils and has its administrative capital at Ajumako. Ghana's 2021 Population and Housing Census reported the population of the District as being predominantly rural, with 65% of its 120,586 inhabitants living in rural communities [16]. The population, with a density of 217.9 persons per square kilometre, comprises 57,261 men and 63,325 women [16]. The average household size, as recorded in the 2021 census, is 3.1 persons.

The District has a moist semi-equatorial climate, with annual rainfall ranging between 120 and 150 millimetres [15]. The rainfall has a double maxima, with peaks in May–June and September–October. August is the coldest month, having a mean monthly temperature of 26 °C, while March–April records the highest mean monthly temperature of 30 °C [15]. The soils have a variable texture, ranging from clayey, sandy, and loamy soils from zone to zone.

With data on sanitation and toilet usage collected during the 2021 Population and Housing Census yet to be published, the latest data are from the 2010 Population and Housing Census [17], which indicates that most of the inhabitants (46.2%) rely on public toilets while 37.4% use the various types of pit latrines at home. Some 3% of the population use a water closet while 13% do not have access to any type of sanitation facility.

### 2.2. Study Design

The study was designed to compare the usage and microbial safety of shared and unshared toilets. The microbial safety was determined experimentally, whereas the usage was assessed using a semi-structured questionnaire. The questionnaire was designed to collect data on the household population, number of households using shared toilets, and cleaning regime. Usage of the toilets was compared on the basis of the user populations or the relative access to sanitation offered by the two types of toilets, as reported by the owners. The microbial safety of the toilets was compared on the basis of the faecal microbial presence (prevalence) and load (concentration) on two common contact surfaces of the toilets, namely the toilet seat and door handles. Faecal contamination was detected by the presence of *E. coli* and/or other faecal coliforms. The actual load of *E. coli* was enumerated and reported separately from that of other faecal coliforms.

### 2.3. Sample Size and Sampling Approach

Due to the low coverage of household toilets and the inclusion criteria adopted for the study, a non-probabilistic sampling approach was used. Based on the study design described above, toilets included in the study were those that had a seat, rather than a squat hole, as the user interface, and that were also fitted with a door. To avoid the confounding effect of technology types within the two groups of toilets (shared and unshared), only ventilated improved pit latrines, which is the predominant technology type, were used for the study. Toilets satisfying the above inclusion criteria were literally searched in twelve communities within five of the nine Area Councils, namely Abaasa, Enyan Denkyira, Essiam, Etsi Sonkwa, and Mando. Budgetary and logistical constraints allowed a maximum of 100 toilets to be targeted, with equal proportions of shared and unshared facilities. However, actual availability allowed a total of 48 shared and 51 unshared, for a total of 99 toilets to be sampled for the study. For each toilet, swab samples were taken from the seat and door handles for assessment of the microbial load. In addition, the landlord (owner of the toilet) or a well-informed adult representative above the age of 18 was selected to respond to a semi-structured questionnaire designed to collect data on the usage and user population of the toilet.

### 2.4. Microbial Sample Collection, Preservation and Analysis

Samples were collected using the method/procedure described by Kwetché et al. [18]. Using sterile swab sticks, the sampling from each surface was carried out by primarily dipping the cotton bud of the sterile swab into a sterile (0.9% NaCl) physiological saline solution. Two different swabs were rubbed separately on the surface of the toilet seat and handle of the door in a prescribed pattern, as specified by Public Health England [19]. Streaks were created over the entire surface area with enough pressure to optimize the rubbing. To avoid cross-contamination, used swabs were immediately returned into the sterile swab container before labelling and proper identification. To ensure sample integrity, all labelled samples were kept in an ice chest containing ice cubes to maintain a $-4\,^\circ$C environment and transported to the Environmental Quality Laboratory of the Department of Water and Sanitation, University of Cape Coast.

Samples and/or swab sticks were put into test tubes containing 9 mL peptone water and incubated at 37 $^\circ$C $\pm$ 2 $^\circ$C overnight to ensure the growth and multiplication of swabbed organisms. The peptone water was prepared according to the instructions of the manufacturer (Oxoid Limited, Hampshire, UK) and autoclaved at 121 $^\circ$C at 15 psi for 15 min. A four- to five-step serial dilution was performed on each sample depending on the turbidity of the peptone water after incubation. Eosin methyl blue (EMB) agar, pipet tips, Petri dishes, and all other items used were first autoclaved at 121 $^\circ$C for 15 min at 15 psi. EMB agar was also prepared according Oxoid Limited's instructions. All prepared media were removed from the refrigerator and allowed to attain room temperature prior to use. A total of 0.1 mL of the serially diluted sample was inoculated into the Petri dish following the pour plate method [20]. Samples were incubated at 37 $^\circ$C $\pm$ 2 $^\circ$C for 24–48 h by turning Petri

plates upside down. The microbiological incubator with model number VWR INCU-Line IL 53, manufactured by VWR International BVBA of Belgium, performed a self-calibration on each start-up in addition to a mandatory periodic calibration undertaken by the Ghana Standards Authority.

To identify faecal organisms, the isolated organism was identified based on its morphological characteristics on the EMB agar. *E. coli* exhibits a green metallic sheen/blue-black bull's eye and this organism was confirmed using the indole test [21]. All other faecal organisms were counted together. Colony-forming units for each swab taken were calculated using Equation (1) [22]:

$$CFU/\text{mL} = \frac{(No.\ of\ colonies \times dilution\ factor)}{volume\ plated} \tag{1}$$

*2.5. Data Analysis*

The data were analysed to assess the prevalence and actual quantities/concentrations of faecal contamination on the common contact surfaces in terms of the presence of *E. coli* as a specific faecal coliform and other faecal coliforms (other than *E. coli*) present. The prevalence (proportions) of toilets that had door handles and toilet seats testing positive for these indicator organisms was analysed. The prevalence of faecal contamination within the two groups of toilets was compared using odds ratios computed with the aid of the MedCalc online statistical calculator [23], which is based on Altman [24]. The samples were also analysed for the actual concentrations/quantities of the indicator organisms present on the contact surfaces. For each sample, the average of the microbial analyses results obtained in triplicate was calculated to represent the sample. The data were checked and found to be non-normally distributed. Hence, a non-parametric statistical method, specifically, the Mann–Whitney U test, was used in the comparison of the concentrations of the indicator organisms between the two groups of toilets. This was carried out with the aid of the SPSS Statistical Software. The z-score, which is a normal approximation of the Mann–Whitney U statistic and the corresponding *p*-values, as calculated by the SPSS Software, is reported alongside the median concentrations, the mean ranks, and the sum of ranks.

## 3. Results and Discussion

*3.1. Statistical Analysis of the Usage of Shared and Unshared Toilets*

Table 1 presents the results regarding the number of households and persons that used the shared and unshared toilets.

**Table 1.** Statistics of users of shared and unshared toilets.

| Statistic | Frequency (% within Sharing Status) | |
|---|---|---|
| | **Shared (*n* = 48)** | **Unshared (*n* = 51)** |
| Number of households sharing a toilet | | |
| One household | - | 51 (100%) |
| Two households | 46 (95.83%) | - |
| Three households | - | - |
| Four households | 2 (4.17%) | - |
| Total | 48 (100%) | 51 |
| Average | 2.08 | 1.00 |
| Number of persons using a toilet | | |
| Minimum | 5 | 2 |
| Maximum | 14 | 6 |
| Average | 7.02 | 3.69 |

Toilet sharing mostly occurred between two households. From the average user populations indicated in Table 1, it can be seen that a shared toilet offered access to sanitation to about 90% more people than an unshared one. The level of sharing among the toilets in this study area falls well within the recommended benchmark proposed by the Sanitation Task Team to the JMP to be considered for inclusion in the normative basic definition for basic sanitation, as mentioned earlier.

The average user population of single-household toilets reasonably reflects the average household size in Ghana (3.6) but is higher than that of the Ajumako–Enyan–Essiam District itself (3.1), as revealed by the 2021 Population and Housing Census [16]. Similarly, even though nearly all shared toilets were shared by two households, their average user population (7) is higher than two times the average household size of the District (6.2). The disparity between the user population of the toilets and the actual average household size of the District could be attributed to the non-probabilistic sampling approach adopted in this study. However, compared to the average household size in sub-Saharan Africa (6.9) and the global average of 4.9 persons [25], the toilets in this study were used by a relatively lower number of persons per household. Even though they are shared, the number of people sharing could qualify them as unshared toilets in some countries of sub-Saharan Africa.

Compared to the average household sizes in sub-Saharan Africa and the world, the largest number of persons using a shared toilet [13] is equivalent to two households in sub-Saharan Africa and three households globally. A search through the literature did not reveal any international guidelines on the optimum number of persons expected to use one household toilet, other than the STT's suggested benchmark of 30, which was not adopted by the JMP [6]. For emergency situations, a maximum usage rate of 20 persons per toilet is recommended [26]. In Ghana, the Community Water and Sanitation Agency's (CWSA) sector guidelines for the design of household toilets in small towns and rural areas specifies a maximum usage rate of 25 persons per toilet [27]. Even for shared toilets, the largest user population in the study location falls within the CWSA design guidelines.

While highlighting the greater access to sanitation offered by latrine sharing, it is important to recognise its potential implications for the accessibility, safety, and hygienic condition of the toilet. A high usage rate has been associated with long queues in the use of shared toilets [28]. The user population of the shared toilets could potentially lead to long queuing time, especially during peak periods such as the rush hours of the morning. Such situations may cause users to resort to open defecation or a public toilet. To assess the possibility of this phenomenon, a question was posed to the respondents to verify whether such tendencies are higher among the users of the shared toilets. To the question "*Over the past 24 h, have you used a public toilet or practised open defecation?*", 4 out of 48 respondents of shared toilets, representing 8.3%, and 8 out of 51 respondents of unshared toilets, representing 15.7%, responded in the affirmative. However, their action may not be directly related to circumstances surrounding the sharing status or conditions of their toilets. This is because, as they explained in a follow-up question, they were simply not at home when they needed to defecate. This implies they may have used a public or institutional toilet while at work or selling at the market or may have practised open defecation or 'dig-and-cover' while working on their farms. It must, however, be noted that the possibility of someone deferring defecation until such time when they are not in the house due to some cause of dissatisfaction with their home latrines cannot be ruled out. Nevertheless, the above data indicate that the odds of a person having access to a shared household toilet resorting to a public toilet or practising open defecation is lower but not significantly different from that of someone having access to an unshared toilet (OR = 0.49; 95% CI: 0.14–1.74; $p$ = 0.270). Another aspect concerning the number of persons using the two categories of toilets is how this affects the hygiene and, for that matter, the microbial safety of the toilets. This is examined in detail in the next section.

### 3.2. Microbial Safety of Shared and Unshared Toilets

3.2.1. Prevalence of Faecal Contamination among Toilets

Table 2 presents an overview of the proportions of toilets with faecal contamination on common contact surfaces, assessed by the presence of *E. coli* and any other faecal coliform. It also shows how the proportions among the shared toilets compare to those in the unshared toilets in terms of odds ratios.

**Table 2.** Proportions of toilets with faecal contamination on common surfaces.

| Contact Surface | Indicator Organism | Frequency (% of Detection) | | | OR (95% CI) |
|---|---|---|---|---|---|
| | | All Toilets (*n* = 99) | Shared (*n* = 48) | Unshared (*n* = 51) | |
| Door handle | *E. coli* | 87 (87.88%) | 43 (89.58%) | 44 (86.27%) | 1.37 (0.40–4.64) |
| | Other FCs | 74 (74.75%) | 31 (64.58%) | 43 (84.31%) | 0.34 (0.13–0.89) * |
| | *E. coli* or other FCs [1] | 91 (91.92%) | 44 (91.67%) | 47 (92.16%) | 0.94 (0.22–3.97) |
| Seat | *E. coli* | 92 (92.93%) | 45 (93.75%) | 47 (92.16%) | 1.28 (0.27–6.03) |
| | Other FCs | 83 (83.84%) | 40 (83.33%) | 43 (84.31%) | 0.93 (0.32–2.71) |
| | *E. coli* or other FCs | 95 (95.96%) | 47 (97.92%) | 48 (94.12%) | 2.94 (0.29–29.26) |
| Either door handle or seat [2] | *E. coli* | 97 (97.98%) | 47 (97.92%) | 50 (98.04%) | 0.94 (0.06–15.46) |
| | Other FCs | 91 (91.92%) | 42 (87.50) | 49 (96.08) | 0.29 (0.05–1.49) |
| | *E. coli* or other FCs | 98 (98.99%) | 47 (97.92%) | 51 (100%) | 0.31 (0.01–7.73) |

\* = Significant at 5% confidence level; FCs = faecal coliforms; [1] Either *E. coli* or other FCs present; [2] The specified indicator organism(s) present on either door handle or toilet seat.

Generally, the prevalence of faecal contamination of the common contact surfaces was very high among the toilets. Almost all the toilets (98 out of 99) had either the seat or door handle testing positive for either *E. coli* or other faecal coliforms. Only one toilet (shared) had no form of faecal contamination on either the seat or door handles. For the door handles, 91 of the toilets (91.9%) had faecal contamination detected by the presence of either *E. coli* or other faecal coliforms. For the toilet seats, the corresponding prevalence was 96%. The faecal contamination of toilet seats could be due to the direct deposition of faeces on the surfaces. The door handles were possibly contaminated through unclean hands. This suspicion is supported by the observation of no functional handwashing facilities located inside or close to the toilet facilities during the fieldwork. Even though the mere presence of a handwashing station may not guarantee usage, the probability of use could be high. The prevalence of *E. coli* on the contact surfaces assessed was higher for unshared than shared toilets, but the differences were not statistically significant. However, the prevalence of other faecal coliforms on the door handles of unshared toilets was significantly higher than that of shared toilets. In Tanzania, Massa et al. [29] found the likelihood of faecal matter inside the shared toilets to be less than that of unshared toilets.

Both shared and unshared toilets could potentially result in the transmission of diseases among the users of the toilets. 'Innocent' persons who may encounter the door handles intentionally or accidently could be at risk of infection. Women are the most vulnerable due to frequent use (for defaecation, urination, and menstrual hygiene management), cleaning of the facilities, caring for the sick and elderly, and the waste disposal of faeces of children, the sick, or the elderly [30]. The risk could be minimized or, at best, eliminated if the users implemented potential risk-reduction measures such as effective handwashing with soap and regular thorough cleaning of the toilets, including the wiping of contact surfaces. The cleaning and disinfection of contaminated surfaces are effective methods to control pathogens [31]. Moreover, wiping the contact surfaces once is reported to result in a $1\log_{10}$ reduction in bacteria concentration and $3 \log_{10}$ when a second wipe is implemented [32]. Similarly, handwashing without soap led to a $1\log_{10}$ reduction in bacteria concentration and this increased to a $1.7 \log_{10}$ reduction when handwashing was implemented with soap [33]. Ramlal et al. [34] tested the effectiveness of potential risk-reduction interventions on *E. coli* concentration on the contact surfaces of community ablution blocks and found that the wiping of surfaces (at least twice prior to contact) and washing of hands with soap have the potential to significantly reduce the risk of infection.

There is an urgent need for the Environmental Health and Sanitation Department of the local District Assembly to educate the users on the need to thoroughly clean their toilets, proper ways of cleaning (including the cleaning of contact surfaces), and personal hygiene practices (handwashing with soap after using the toilet).

3.2.2. Concentrations of Faecal Coliforms on Contact Surfaces

Table 3 presents the microbial load on the sampled contact surfaces. Due to the non-normal distribution of the data, the comparison of the concentrations between the shared and unshared toilets was carried out with non-parametric statistical tools. Hence, the median CFU/mL of the indicator organisms is reported (instead of the mean) alongside the mean ranks and sum of ranks.

**Table 3.** Levels of faecal contamination of common contact surfaces.

| Contact Surface | Indicator Organism | Shared | Unshared | Z-Score (*p*-Value) |
|---|---|---|---|---|
| Door handle | *E. coli*<br>Median CFU/mL × 105<br>Mean rank<br>Sum of ranks | <br>34.333<br>50.14<br>2406.50 | <br>54.667<br>49.87<br>2543.50 | 0.046 (0.964) |
| | Other FCs<br>Median CFU × 105/mL<br>Mean rank<br>Sum of ranks | <br>7.000<br>43.47<br>2086.50 | <br>16.333<br>56.15<br>2863.50 | 2.213 (0.027) * |
| Toilet seat | *E. coli*<br>Median CFU/mL × 105<br>Mean rank<br>Sum of ranks | <br>103.167<br>47.32<br>2271.50 | <br>125.000<br>52.52<br>2678.50 | 0.900 (0.368) |
| | Other FCs<br>Median CFU/mL × 105<br>Mean rank<br>Sum of ranks | <br>24.000<br>49.81<br>2391.00 | <br>31.667<br>50.18<br>2559.00 | 0.063 (0.950) |

* = Significant at 5% confidence level; FCs = faecal coliforms.

Faecal contamination was higher on toilet seats than door handles irrespective of the sharing status. It can be deduced from Table 3 that *E. coli* concentrations on the door handles of shared and unshared toilets were 33% and 44% of the concentration on the toilet seats, respectively. Contrary to our findings, Ramlal et al. [34] found the *E. coli* concentration on door handles to be significantly higher than toilet seats in community ablution blocks located in informal settlements in South Africa. The cleaning regime may have contributed to this divergence. Community ablution blocks are managed by caretakers who are responsible for cleaning, as opposed to the cleaning of the toilets by users in the current study community. It is possible that the cleaners of the community ablutions regularly cleaned the toilet seats but paid no attention to the door handles.

Table 3 shows that a similar comparison exists between the shared and unshared toilets in terms of the microbial loads, as in the case of the prevalence (proportions of toilets with faecal contamination). Practically, the prevalence of faecal contamination and the actual microbial loads showed no difference between shared and unshared toilets. Even though the level of other faecal coliforms (excluding *E. coli*) on toilet door handles was significantly lower among shared toilets both in terms of the prevalence (OR = 0.34; 95% CI: 0.13–0.89) and microbial load (Z = 2.213; *p* = 0.027), the overall faecal contamination (including *E. coli*) was comparable between the two groups. The outcome of this study confirms the findings from Tanzania, where points of hand contact in shared toilets were found to be significantly less contaminated with *E. coli* than unshared toilets (9 vs. 18 *E. coli*/100 mL, *p* = 0.04) [13]. The results from this study suggest that the microbial safety of the toilets may be a reflection of the general toilet usage and hygiene practices of the residents of the study communities, rather than the consequence of toilet sharing.

The loads of the indicator organisms on both door handles and toilet seats are relatively lower on the shared toilets than the unshared ones, even though the difference is not

statistically significant in most scenarios of surface and indicator types. The results show that shared toilets could be managed to be even safer than unshared ones. The practical question is '*why are the shared toilets not necessarily less safe in spite of their significantly higher user populations?*'. A number of reasons could account for this. Firstly, when toilets are shared by different households, users become more conscious of the hygienic condition of their toilets for fear of contracting diseases from the other users. Secondly, users of shared toilets are able to mobilize resources to clean and maintain the toilets. Hailu et al. [35] identified low monthly household income as a barrier to the cleaning of shared toilets. The cost associated with the cleaning of toilets does not become a burden for a single household when toilets are shared. Finally, when toilets are shared by households who share some bond, the users are able to activate their spirit of social cohesion to clean the toilets. Sharing cleaning responsibilities among users of shared toilets could result in frequent cleaning. The relationship and cooperation among users, and a commitment to cleaning have been reported as key determinants for the cleanliness of shared toilets [36–38]. In Ghana and Kenya, Antwi-Agyei et al. [39] found no statistically significant association between a toilet's cleanliness and the number of households sharing it, even though clean toilets were used by relatively fewer households than dirty toilets. Many of the studies that associate unclean toilets with sharing have been driven by communal and public toilets and not necessarily the household-level sharing of toilets [11].

The prevalence and concentrations of *E. coli* were higher than those of all other faecal coliforms combined. This implies (or confirms the assertion) that *E. coli* is a more reliable indicator organism for faecal pollution. In other words, it is less likely that faecal pollution will fail to be detected when *E. coli* is used as the indicator organism because of their prevalence in human and animal faeces compared to other thermotolerant coliforms. Fast, sensitive, and easy-to-perform detection methods for *E. coli* are readily available and affordable [40].

## 4. Conclusions

From the results of the study, it can be concluded that the sharing of toilets at the household level has the potential to increase toilet access by nearly twofold without necessarily making the toilets less safe, as compared to those used by single households within the same socio-cultural settings. Even though the toilets sampled had a high prevalence of faecal contamination on common contact surfaces and could serve as a medium of disease transmission, the low safety of the toilets was independent of their sharing status and may rather reflect the general user behaviour and hygiene practices of the residents of the study area. The prevalence of faecal contamination among the toilets and the actual microbial load of indicator organisms were more promising (relatively lower) in the shared toilets than the unshared ones, even though the differences were not significant in most scenarios featuring different types of contact surface and indicator organism. The findings of this study give credence to calls to disaggregate toilets shared at the household level by a few households or some specified number of users from public or communal toilets, and to accord household-level shared toilets some recognition in the monitoring of progress towards safe sanitation. This study also highlights the importance of hygienic usage and management practices among toilet users irrespective of the sharing status of the toilet. More importantly, the high prevalence of faecal contamination of the common contact surfaces underscores the need for hand washing with soap after using a toilet facility and the wiping of door handles as part of the cleaning regime. It is imperative for the authorities of the local District Assembly to intensify health and hygiene education among the residents of the study communities as an essential complement to sanitation access.

**Author Contributions:** Conceptualization, P.A.O. (Peter Appiah Obeng), E.A. and P.A.O. (Panin Asirifua Obeng); methodology, P.A.O. (Peter Appiah Obeng), E.A., P.A.O. (Panin Asirifua Obeng), M.O.-P. and A.K.M.; validation, A.B. and S.A.Q.; formal analysis, P.A.O. (Peter Appiah Obeng), E.A. and M.O.-P.; investigation, P.A.O. (Panin Asirifua Obeng), A.K.M. and S.A.Q.; resources, P.A.O. (Peter Appiah Obeng), E.A., P.A.O. (Panin Asirifua Obeng) and M.O.-P.; data curation, P.A.O. (Peter Appiah Obeng), E.A., M.O.-P. and A.B.; writing—original draft preparation, P.A.O. (Peter Appiah Obeng) and E.A.; writing—review and editing, P.A.O. (Panin Asirifua Obeng), M.O.-P., A.K.M., A.B. and S.A.Q.; visualization, P.A.O. (Peter Appiah Obeng) and E.A.; supervision, P.A.O. (Peter Appiah Obeng), E.A., P.A.O. (Panin Asirifua Obeng) and S.A.Q.; funding, A.B. All authors have read and agreed to the published version of the manuscript.

**Funding:** The APC was funded by the University of Bologna, Italy.

**Institutional Review Board Statement:** Not applicable.

**Informed Consent Statement:** Not applicable.

**Data Availability Statement:** Data are contained within the article.

**Acknowledgments:** The authors are grateful to Samuel Nketsia and the other laboratory staff of the Department of Water and Sanitation, University of Cape Coast, and Emmanuel Kekeli Akuaku (Cape Coast Technical University) for their support during the sampling and laboratory analysis. The authors are also grateful to the academic editor, editorial team, and anonymous reviewers for their valuable comments and input.

**Conflicts of Interest:** The authors declare no conflict of interest.

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
