# Peer review of "Usage and Microbial Safety of Shared and Unshared Excreta Disposal Facilities in Developing Countries: The Case of a Ghanaian Rural District"

_sustainability, doi:10.3390/su151310282_

Round 1

Reviewer 1 Report

The scientific name should be in italics. Check and correct it throughout the text.
The introduction is too long. It should be comprehensive, logical, and to the point.
Lines 81–108 should be part of the discussion.
ml should be mL. Check and correct it throughout the text.
Write a brief description of a semi-structured questionnaire.
There should be no repetition of results in the text or in the table.
Discussion should be free of results. Do not waste entire sentences restating your results.
Table 2 is confusing. The row "E. coli or other FCs" indicates what? Why did you add "Either door handle or seat?" What do you want to explain in these lines?

Extensive editing of the English language is required

Reviewer 2 Report

The study discussed and compared the usage and microbial safety of shared and unshared excreta disposal facilities in Ghanaian rural district. The study also investigated association between their microbial safety and sharing status using a semi-structured questionnaire data collection in addition to assessment of faecal contamination following standard swab sampling and analytical protocols. The results found that sharing toilets affords about 90% more household-level access to sanitation and there was a high prevalence of faecal contamination of the door handles and seats of both toilets and concluded that the sharing of toilets at the household level nearly doubles access to sanitation at home without necessarily exposing the users to a higher risk of faecal-oral disease transmission.

The main points here the methodology and conclusion as they both need more improvement, in addition to and a little language editing to whole manuscript.  

Prevalence of faecal contamination among toilets is not enough, we need the whole profile of screening of other bacteria (gram negative and gram positive).

Quality control in is not clear in methodology, author need to specify quality control steps like instrument calibration, ATCC cultures etc.

Quality of English language needs minor revision.

Reviewer 3 Report

Dear authors,

I consider the manuscript "Usage and Microbial Safety of Sheared and Unsheared ExcretaDisposal Facilities in Developing Countries", I think that it is an important contribution presented adequately described with scientific rigor. The manuscript contain an adequate and well documented discussion. Inside the manuscript you will find suggestions and typographic errors detected to be consider.

Best regards

Round 2

Reviewer 1 Report

Accept in present form

Minor editing of English language required